# New Mutations in *HFE2* and *TFR2* Genes Causing Non *HFE*-Related Hereditary Hemochromatosis

**DOI:** 10.3390/genes12121980

**Published:** 2021-12-13

**Authors:** Gonzalo Hernández, Xenia Ferrer-Cortès, Veronica Venturi, Melina Musri, Martin Floor Pilquil, Pau Marc Muñoz Torres, Ines Hernandez Rodríguez, Maria Àngels Ruiz Mínguez, Nicholas J. Kelleher, Sara Pelucchi, Alberto Piperno, Esther Plensa Alberca, Georgina Gener Ricós, Eloi Cañamero Giró, Santiago Pérez-Montero, Cristian Tornador, Jordi Villà-Freixa, Mayka Sánchez

**Affiliations:** 1Iron Metabolism: Regulation and Diseases Group, Department of Basic Sciences, Universitat Internacional de Catalunya (UIC), 08195 Sant Cugat del Vallès, Spain; ghernandezv@uic.es (G.H.); xferrerc@uic.es (X.F.-C.); vventuri@uic.es (V.V.); 2BloodGenetics S.L., Diagnostics in Inherited Blood Diseases, 08950 Esplugues de Llobregat, Spain; mmusri@bloodgenetics.com (M.M.); sperez@bloodgenetics.com (S.P.-M.); Ctornador@bloodgenetics.com (C.T.); 3Department of Basic Sciences, Faculty of Medicine and Health Sciences, Universitat Internacional de Catalunya, 08195 Sant Cugat del Vallès, Spain; mfloor@uic.cat (M.F.P.); pmunoz@uic.es (P.M.M.T.); jordi.villa@uvic.cat (J.V.-F.); 4Department of Biosciences, Faculty of Sciences and Technology, Universitat de Vic—Universitat Central de Catalunya, 08500 Vic, Spain; 5Hospital Germans Trias i Pujol, 08916 Badalona, Spain; agnesrh@iconcologia.net; 6Department of Laboratory Medicine/Fundació Hospital de l’Esperit Sant, 08923 Santa Coloma de Gramenet, Spain; aruiz@fhes.cat; 7Hematologia Clinica, Institut Català d’Oncologia, 17007 Girona, Spain; nkelleher@iconcologia.net; 8Department of Medicine and Surgery, University of Milano-Bicocca, 20126 Monza, Italy; sara.pelucchi@unimib.it (S.P.); alberto.piperno@unimib.it (A.P.); 9Medical Genetics—ASST-Monza, S. Gerardo Hospital, 20900 Monza, Italy; 10Centre for Rare Diseases—Disorders of Iron Metabolism—ASST-Monza, San Gerardo Hospital, 20900 Monza, Italy; 11Hematologia i Hemoteràpia, Consorci Sanitari del Maresme, Institut Català d’Oncologia, 08304 Mataró, Spain; mplensa@csdm.cat (E.P.A.); ggener@iconcologia.net (G.G.R.); eloi.canamero@gmail.com (E.C.G.)

**Keywords:** HFE related hereditary hemochromatosis, non-HFE related hereditary hemochromatosis, iron overload, missense, nonsense, homozygous, *TFR2* gene, *HFE2* gene

## Abstract

Hereditary hemochromatosis (HH) is an iron metabolism disease clinically characterized by excessive iron deposition in parenchymal organs such as liver, heart, pancreas, and joints. It is caused by mutations in at least five different genes. HFE hemochromatosis is the most common type of hemochromatosis, while non-HFE related hemochromatosis are rare cases. Here, we describe six new patients of non-HFE related HH from five different families. Two families (Family 1 and 2) have novel nonsense mutations in the *HFE2* gene have novel nonsense mutations (p.Arg63Ter and Asp36ThrfsTer96). Three families have mutations in the *TFR2* gene, one case has one previously unreported mutation (Family A—p.Asp680Tyr) and two cases have known pathogenic mutations (Family B and D—p.Trp781Ter and p.Gln672Ter respectively). Clinical, biochemical, and genetic data are discussed in all these cases. These rare cases of non-HFE related hereditary hemochromatosis highlight the importance of an earlier molecular diagnosis in a specialized center to prevent serious clinical complications.

## 1. Introduction

Iron is a bio-mineral and a micronutrient with a key role in a wide range of cellular processes such as erythropoiesis, oxygen transport (as a critical component of hemoglobin), DNA synthesis or energy production. Since both low and high levels of iron are linked to life threatening complications [1], a complex system of proteins and gene regulation exists to keep iron levels in a safe range and to ensure all the iron-related processes working in optimal conditions. A key regulator of iron metabolism is the hepcidin peptide, the liver-produced iron regulatory hormone coded by the *HAMP* gene [2]. In healthy individuals (Figure 1, left panel), with adequate iron levels in hepatocytes, hepcidin expression is induced by a signaling cascade in which participates several proteins. One of the events in this cascade is the binding of the transferrin protein (TF) with the Transferrin Receptor 1 (TFR1) protein (responsible for the cellular iron uptake) and, with a much-reduced affinity, to the Transferrin Receptor 2 (TFR2) protein (responsible for the iron levels sensing). TFR2, encoded by the *TFR2* gene, is a TFR1-homologue protein that exists as a transmembrane homodimer being mainly expressed in the liver [3] and in bone marrow erythroblasts, where it regulates the activity of the EPO receptor to adapt the production of RBC to the available amount of iron-bound TF [4]. The TF-TFR1 binding displaces the HFE protein that then is free to interact with TFR2 to generate the signal required for hepcidin production. In addition, the Bone Morphogenic Protein (BMP) cytokines BMP2 and BMP6, secreted by the liver sinusoidal endothelial cells (LSECs), interact with the BMP receptors 1 and 2 and the BMP co-receptor HJV protein (a GPI-linked protein encoded by the *HFE2* gene), to also signal for hepcidin production [5]. The BMPR-HJV cascade can be inhibited by the action of the matriptase-2 protein (encoded by the *TMPRSS6* gene). Both signaling events (HFE-TFR2 and HJV-BMPR) trigger the phosphorylation of the SMAD1/5/8 proteins forming a complex with SMAD4 that will translocate to the nucleus and induce the expression of the *HAMP* gene. The hepcidin produced is secreted to the blood stream and will inhibit (by inducing its degradation) the FPN protein, an iron pump that is responsible for the iron absorption in the duodenum and iron release from macrophages. By reducing the FPN levels, hepcidin reduces the iron availability from dietary iron absorption and iron secretion by reticuloendothelial system (RES) macrophages. The relevant function of all these proteins in iron homeostasis has been revealed by the generation on *knock-out* mouse models that phenocopy the human HFE and non-HFE related hereditary hemochromatosis disease [6,7,8,9,10,11].

Alterations in this signaling cascade leads to reduced production of hepcidin and to the manifestations of the different forms of hereditary hemochromatosis (HH) [12]. This reduction in hepcidin production (Figure 1, right panel) results in an excessive iron absorption that eventually accumulates in organs such as liver, heart, joints and pancreas [13]. The iron accumulation in those organs over time produces tissue damage and other clinical manifestations such as cirrhosis, diabetes mellitus, arthropathy, cardiomyopathy, hypogonadism, impotence, and hepatocellular carcinoma. To avoid organ damage, it is vital an early diagnose and early implementation of iron chelation or iron reduction treatment to diminish iron overload.

HH has been traditionally subdivided into four different types associated with mutations in five different genes involved, as seen before, in hepcidin regulation. Type 1 is the most common form of HH (OMIM #235200), and it is linked to mutations in the major histocompatibility complex class I-like protein encoded by the *HFE* gene (OMIM *613609) [14]. Juvenile hemochromatosis is the name of the HH types 2a and 2b (OMIM #602390 and #613313) that are caused by mutations in the hemojuvelin (HJV) protein (coded by the *HFE2* gene) and in the iron hormone hepcidin (coded by the *HAMP* gene) (OMIM *608374 and *606464) [2,15]. Type 3 HH (OMIM #604250) is associated with genetic alterations in the Transferrin receptor 2 (*TFR2*) gene (OMIM *604720) [16]. Type 3 HH was originally considered to be a disease with an adult onset; however, there are reports of patients with a juvenile onset of this disease [17,18,19]. In type 4b HH (OMIM #606069), there are gain-of-function mutations in the iron exporter ferroportin (FPN) protein (coded by the *SLC40A1* gene) (OMIM *604653) [20]. In general, HH is mainly an autosomal recessive disease except for the HH type 4b caused by dominant mutations in the *SLC40A1* gene. Clinically, HH types 2a, 2b and 3 present with more severe and earlier iron overload complications compared with type 1 and 4b.

Recently, a new classification for HH has been proposed, where the previously described types of HH have been reorganized into the “*HFE-related*” HH caused by mutations in the *HFE* gene, the “*non HFE-related*” due to mutations in the *HFE2*, *HAMP*, *TFR2* and gain-of-function mutations in the *SLC40A1* gene, the “*Digenic*” caused by compound heterozygosity between mutations in iron-metabolism-related genes (*HFE* and/or *non-HFE* genes) and lastly, the “*molecularly undefined*” group where the genetic origin is unknown. From this point forward, we will use this new recommended nomenclature [21].

Up to date, 98 patients from 87 different families with non-HFE related HH due to pathogenic mutations in the *HFE2* gene have been described in the literature (Table A1) [22,23]. As we can see in Figure 2, the mutations are scattered across the length of the HJV protein, with no evident clustering.

Up to date, a total of 45 families (66 affected patients) with non-HFE related HH due to pathogenic mutations in the *TFR2* gene have been described in the literature (Table A2) [24,25]. It has been shown that in a hepatic murine system (Hepa1–6 cells), the murine counterparts of four human TFR2 mutations (Met172Lys, AVAQ621_624del, Gln690Pro and Tyr250Ter) cause intracellular retention of the protein at the endoplasmic reticulum [26]. As in HJV, TFR2 mutations are scattered along the protein (Figure 3).

Here, we describe six patients with *non-HFE* related HH from five unrelated families and two previously *unreported mutations in the HFE2 gene and two novel mutations in the TFR2 gene.*

## 2. Materials and Methods

### 2.1. Patients

#### 2.1.1. Non-HFE HH Related Due to Mutations in the *HFE2* Gene: Family 1

Patient 1.II.1 (Figure 4a left) is a male diagnosed at the age of 37. Biochemical data revealed mild basal fasting hyperglycemia with mild insulin resistance (HOMA index of 3.7), vitamin D and folic acid deficit and hypogonadotropic hypogonadism. Hepatic ultrasound showed a slightly increased liver with irregular structure, elastography revealed liver fibrosis and magnetic resonance shows severe liver [T2* 1 ms, corresponding to a liver iron concentration (LIC) of 457 µmol/g dry weight [27] and cardiac (T2* 9 ms) iron overload. Liver biopsy showed cirrhosis and marked iron overload (Figure A1). Despite cardiac overload, echocardiography showed normal morphology and ventricular function with ground glass aspect of the medio-basal ventricular septum. This indicates that T2* was identifying preclinical iron deposition as previously observed in patients with transfusion-dependent thalassemia with cardiac iron overload [28,29]. Because of the severe iron overload, the patient underwent to combined therapy to remove as soon as possible the systemic iron burden. Based on previous experience, we opted for erythroapheresis to maintain isovolemia (considering the cardiac iron overload) plus recombinant erythropoietin administration (8000 UI/week for three weeks/month) to maximize the efficacy of erythroapheresis [30]. In addition, subcutaneous deferoxamine infusion (1 g twice/day for 5 days/week) was used. Iron depletion was achieved after the removal of 15 g of iron in one year. Afterward, phlebotomy therapy every 3–4 months was performed, maintaining ferritin levels around 50 ng/mL.

#### 2.1.2. Non-HFE HH Related Due to Mutations in the *HFE2* Gene: Family 2

Patient 2.II.1 (Figure 4a right) is a male diagnosed at the age of 34 years old. He presented with high serum ferritin levels (but <1000 µg/L) and high serum iron. In addition, he had hypogonadotropic hypogonadism treated with testosterone and moderate hepatic steatosis. As expected for an iron overload disease, the hepcidin levels of the patient were low (0.1919 ng/mL). One year later, serum ferritin levels peaked to 3942 µg/L. Magnetic resonance shows no evidence of iron overload in the heart while in the liver revealed increased iron concentration of 47 µmol/g indicative of hepatic iron overload (normal values <36 µmol/g). Iron chelation with Desferoxamine was used as the main therapeutic treatment. Initially, phlebotomies were performed in combination with iron chelation but had to be stopped due to intolerance. Iron chelation treatment ended in 2020 and the patient is now asymptomatic. The patient will continue with maintenance therapy.

#### 2.1.3. Non-HFE HH Related Due to Mutations in the *TFR2* Gene: Family A

Patients A.II.1 and A.II.2 (Figure 4b upper first panel) are two male brothers of Asian origin diagnosed with HH at 35 and 37 years old respectively. Both presented with high levels of serum ferritin and iron, while in both patients, the hepcidin levels were 0.2395 and 0.0111 ng/mL respectively. Hepatic magnetic resonance showed a severe hepatic iron overload (282.97 µmol Fe/g and 265 µmol Fe/g). The treatment option for both patients consisted of weekly phlebotomies in combination with iron chelation (Desferoxamine). A.II.1 proband started the phlebotomies in January 2019 (weekly) and the Desferoxamine treatment in May 2019. In February 2021, after 100 phlebotomies and approximately 22 g of iron removal the ferritin levels dropped to normal levels, but transferrin saturation remained high. A.II.2 proband started the phlebotomies in July 2017 (once a month) and the Desferoxamine treatment in January 2018 (initial dose of 1080 mg/day that eventually was increased to 2160 mg/day in May 2018). At the last data available (February 2021), the patient accumulated a total of 46 phlebotomies that removed a total amount of 9 g of iron and resulted in the normalization of ferritin and transferrin saturation parameters. The iron depletion was partially confirmed by the last hepatic magnetic resonance performed to the A.II.2 patient that showed a moderate iron overload (70.33 µmol Fe/g).

#### 2.1.4. Non-HFE HH Related Due to Mutations in the *TFR2* Gene: Family B

Patient B.II.1 (Figure 4b upper second panel) is a male of 46 years old diagnosed in 2012 with hemochromatosis that presented with hyperferritinemia and severe hepatic iron accumulation (300 µmol Fe/g) detected by hepatic magnetic resonance. The patient also suffers from dyslipidemia and internal hemorrhoids. The patient does not consume alcohol and is an ex-smoker as of May 2014. Genetic analysis shows that this patient is a carrier for the Cys282Tyr mutation in the HFE gene. Secondary to the hemochromatosis, the patient presents with severe chronic arthropathy in feet, spine (spondylarthrosis) and hands. The treatment initially was monthly erythroapheresis (later, the rate of erythroapheresis was reduced to once every two months). In January 2015, phlebotomies were introduced as part of the treatment. In May 2017, the hepatic magnetic resonance showed no sign of hepatic iron overload.

#### 2.1.5. Non-HFE HH Related Due to Mutations in the *TFR2* Gene: Family C

Patient C.II.1 (Figure 4b bottom panel) is a male diagnosed at age 25 whose parents are consanguineous (first cousins). He presented with high serum ferritin and high transferrin saturation compatible with HH. The hepatic magnetic resonance showed iron overload (123.9 µmol Fe/g). In February 2021, phlebotomies were started at a rate of once a week. After 12 rounds of phlebotomies (April 2021), the ferritin levels and transferrin saturation dropped to 377 ng/mL and 13.9% respectively. At this point, the frequency of phlebotomies was reduced to once a month. At the last control (July 2021), the ferritin and saturation dropped to normal levels (16 ng/mL and 4.8% respectively). Currently, the phlebotomies have been stopped. Iron depletion was reached after 15 rounds of phlebotomies that removed approximately 3 g of iron. Both proband and its sister also suffer from sensorineural hearing loss.

### 2.2. DNA Sequencing and Analysis

Patients were diagnosed using a targeted NGS gene panel (v16) for HH (BloodGenetics #10010 panel) including the following known genes involved in Hereditary Hemochromatosis: *BMP6, FTH1, FTL, GNPAT, HAMP, HFE, HFE2, PIGA, TFR2* and *SLC40A1* and methodologically as reported previously by Vila-Cuenca [31]. Variants were validated by conventional Sanger. Data analysis was performed using VarSome clinical software [32]. The CADD score was calculated using the web server (https://cadd.gs.washington.edu/snv accessed on 15 September 2021). Reported mutations in this study have been submitted to ClinVar (http://www.ncbi.nlm.nih.gov/clinvar, accessed on 1 December 2021).

### 2.3. Hepcidin Determination

Hepcidin concentrations in patients’ plasma/serum samples were quantified by competition enzyme-linked immunoassay (C-ELISA) using the hepcidin-25 (human) enzyme immunoassay kit (DRG) according to the manufacturer’s protocol. Samples and standards were run in duplicate. Hepcidin concentrations were interpolated from standard curves generated by a four-parameter logistic nonlinear regression model using Prism (GraphPad Software Inc., La Jolla, CA, USA).

### 2.4. Computational Studies

#### 2.4.1. Computational Model of TFR2 and TFR2/TF Based on Comparative Modeling with TFR1 Crystal Structure

The human TFR2 sequence was obtained from the UniProt database with ID Q9UP52. Using the available crystal structures for the TFR1 protein (PDB IDs: 1CX8, 1DE4, 1SUV, 2NSU, 3KAS, 3S9L, 3S9M, 3S9N, 6D03, 6D04, 6D05, 6OKD, 6W3H, and 6Y76), one thousand comparative models were generated for the TFR2 protein using the Modeller program [33]. The structures were further clustered into one hundred RMSD clusters and were subsequently docked to the TF protein based on the available complex of TFR1 and TF (PDB 1SUV). Each structure was optimized several times by searching the local conformational space using the Rosetta modeling suite [34]. The lowest energy model produced was then selected as the TFR2/TF heterodimer model. To build the dimeric model of TFR2 interacting with TF, we docked the previously obtained model with a copy of itself, based on the relative orientation of the TFR1 dimer in the 1SUV structure. The docked pose was subjected to several trajectories of full energy optimization, and the lowest energy score was selected as the final hetero-tetrameric model. The TFR2/TF D680Y mutant model was derived from the previous structure by optimizing the rotameric packing of residues surrounding position 680. Different distance thresholds were used to select which residues were allowed to repack during this optimization. Likewise, the lowest energy structure was selected as the final dimeric model.

#### 2.4.2. Sequence Similarity Network of TFR1 and TFR2 Orthologs

Orthologous sequences for TFR1 and TFR2 were downloaded from the NCBI database [35]. To remove highly similar sequences, they were filtered for a maximum of 95% of sequence identity using the CD-HIT program [36]. A pairwise sequence similarity matrix was calculated among the selected protein sequences using the MAFFT program [37], and interaction edges were defined for pairwise similarities above a threshold of 59%. The final network was calculated and depicted with the Cytoscape program using an organic layout [38]. The NCBI codes employed to build the protein sequence similarity network are available upon request.

## 3. Results

### 3.1. Novel Cases of Non-HFE Related HH

Clinical data for each case is present in the patients and methods section; probands presented high serum iron, high serum ferritin, and high transferrin saturation levels, as expected for HH patients, and with low hepcidin levels when available. The characteristic liver iron overload typical of HH was confirmed by liver MRI in the 2.II.2, A.II.1, A.II.2 and C.II.1 probands (Table 1 and Table 2) or by liver biopsy (1.II.1 proband), while some of the probands presented with other HH-associated complications such as hypogonadotropic hypogonadism, arthropathy and diabetes mellitus. Extended biochemical data is depicted in Table 1 for non-HFE patients with HFE2 mutations and Table 2 for non-HFE HH patients with *TFR2* mutations.

### 3.2. Identification of New Mutations Associated with Non-HFE Related HH

#### 3.2.1. Mutations in the *HFE2* Gene

In proband 1.II.1 (Figure 4a left), a NGS targeted diagnostic panel identified in the *HFE2* gene a novel single base deletion in exon 3 (c.445delG) introducing a frameshift mutation (p.Asp149ThrfsTer97) in a homozygous state. Parents were both heterozygous for the same mutation. The parents of the proband denied consanguinity, but they both originated from the same small village in South Italy. This single base deletion generates a truncated HJV product containing the first 148 correct amino acids plus 97 incorrect residues before a premature Stop codon. This variant has an rs code (rs1553769690) with a low frequency (MAF = 0.000004). VarSome Clinical software classifies this mutation as likely pathogenic according to ACMG rules (PVS1 and PM2). HFE testing for Cys282Tyr and His63Asp variants was negative.

Targeted NGS sequencing of the 2.II.2 proband (Figure 4a right) showed a novel nonsense variant in the *HFE2* gene in homozygous state. This variant (c.187C > T) changes an arginine codon to a stop codon (p.Arg63Ter) that results in a truncated HJV protein. This variant is novel and has no rs code. No genetic information is available for the rest of the family. VarSome Clinical software classifies this mutation as pathogenic according to ACMG rules (PVS1, PP3 and PM2). No other variants of interest were found in any of the HH-related genes.

Both newly discovered mutations in the *HFE2* gene produce a truncated HJV protein that clearly would be non-functional, explaining the genetic cause of the non-HFE related HH phenotype in these probands.

#### 3.2.2. Mutations in the *TFR2* Gene

Genetic analysis performed in the A.II.1 and A.II.2 probands (Figure 4b upper first panel) showed that both patients carry the same missense variant in homozygous state in the *TFR2* gene. This previously unpublished variant (c.2038G > T) changes an aspartic acid to a tyrosine (p.Asp680Tyr). This variant has an rs code (rs1302587036) with a low frequency (MAF = 0.0000319). Nineteen out of twenty-five bioinformatic prediction software classified this variant as unknown significance, likely pathogenic (Varsome Clinical software) according to ACMG rules (PM2, PP1, PP3 and BP1). CADD score for Asp680Tyr mutation is 29.8. Computational analysis and modeling are discussed in the Section 3.3. Both probands are homozygous WT for the His63Asp, Ser65Cys and Cys282Tyr HFE mutations.

Regarding the B.II.1 proband, DNA analysis showed a previously reported nonsense mutation in the *TFR2* gene [19]. In our patient, the variant is in homozygous state (Figure 4b upper third panel). This variant (c.2343G > A) changes a tryptophan codon to a stop codon (p.Trp781Ter) that results in a truncated TFR2 protein affecting the dimerization domain of the protein. This variant has an rs code (rs768907730) with a low frequency on the general population (MAF = 0.00000798). Nine out of ten bioinformatics prediction software classified this variant as Pathogenic (Varsome Clinical software) according to ACMG rules (PVS1, PM2, PP3 and PP5). In addition to the TFR2 mutation, our patient also carries the pathogenic p.Cys282Tyr mutant allele in the HFE gene in heterozygous state, which could aggravate the severity of the disease in this patient.

The C.II.2 proband carries a nonsense mutation (p.Gln672Ter, Figure 4b lower panel) that was previously reported in a pediatric patient in heterozygous state in combination with a second nonsense mutation in TFR2 in a Spanish patient [23]. In our patient, the mutation is in a homozygous state. This variant has an rs code (rs1051249273) with a low frequency on the general population (MAF = 0.0000477). This nonsense mutation truncates the TFR2 protein at the beginning of the dimerization domain, clearly affecting the normal function of TFR2. Nine out of ten bioinformatic prediction software classified this variant as pathogenic (Varsome Clinical software) according to ACMG rules (PVS1, PM2 and PP3). No other mutations in HH-related genes were found.

### 3.3. Computational Studies on the TFR2 p.Asp680Tyr Mutation

The variants in the *TFR2* gene found in the B.II.1 and C.II.1 probands are clearly pathogenic since they are nonsense mutations, but the pathogenicity of the missense mutation found in family A is not a straightforward conclusion. To assess the genetic implications of the TFR2 p.Asp680Tyr mutation, a series of protein prediction software and structural modeling was used.

#### 3.3.1. Modeling the Effect of the Asp680Tyr Mutation over TFR2 Activity

To understand the effect of the Asp680Tyr mutation over the TFR2 activity, we built a model of TFR2 complexed with transferrin (TF) based on the available crystallographic structure of the TFR1 dimer bound to TF [39]. Position 680 is located at the C-terminus end of TFR2, in its helical domain, near the TF binding and the dimeric interacting regions (Figure 5A and Figure A3). While human TFR1 and TFR2 share a 43% sequence identity, their amino acid sequences around the mutated region are highly conserved (Figure 5B). Moreover, a comparison between the TFR2/TF dimer model and its crystallographic TFR1/TF counterpart shows a highly similar chemical environment for the TFR2-Asp680 (TFR1-Asp648) residue (Figure 5C,D). Conversely, despite not directly contacting TF, the TFR2-Asp680 sidechain creates a network of interactions that stabilize the side-chain conformation of residue TFR2-Arg683, which participates in a critical saline bond with the TF-Asp356 residue (Figure 5D). Among the residues interacting with TFR2-Asp680, we found residue TFR2-Gln784 (Figure 5D), located at the helix α-18, which participates in an intermolecular TFR2/TFR2 interaction with residue TFR2-Gln340, located at the loop between the β-6 and β-7 strands (Figure 5A).

We modeled the TFR2-Asp680Tyr mutation over the TFR2/TF hetero-tetramer by re-optimizing the rotameric states of surrounding residues using the Rosetta all-atom force field [41]. The mutated structure shows a preference for TFR2-Tyr680 to interact with TFR2-Asn788 via a hydrogen bond (Figure 5E) while keeping hydrophobic interactions with nearby residues that stabilize its side-chain conformation. Thus, the pre-organizing role that TFR2-Asp680 plays over the TRF2-Arg683/TF-Asp356 interaction appears to be abolished when mutated for a tyrosine residue. Previous work analyzed the effect of mutations in the analogous TFR1-Asp648 position over TF binding activity [42]. They showed that a conservative mutation, Asp648Glu, diminished binding to 57% of WT activity; meanwhile, an Asp648Ala substitution reduced it to 16%. Together, this indicates a relevant role for residue Asp680 over the TFR2 binding activity toward TF. While the TFR2-Asp680Tyr mutation could also affect the TFR2/TFR2 dimeric interaction, we expect it to be of a lesser extent than its effect over the TFR2/TF interaction, mainly because TFR2-Asp680 interacts via a hydrogen bond with TFR2-Gln784, an interaction that could be restored with nearby residues by the aid of bridging solvent molecules.

#### 3.3.2. Conservation of TFR2 Position 680 in TFR2 and TFR1 Orthologues

Using a set of 352 orthologous sequences to TFR1 or TFR2, we built a sequence similarity network (SSN) [43] to explore the conservation of the TFR2-680 position among these proteins (Figure A2). In an SSN, nodes represent protein sequences connected by edges to other nodes if their pairwise sequence similarity is above a specified threshold. This particular SSN uses a threshold of 0.59 to define its edges and depicts orthologous sequences to TFR1 (red nodes) and TFR2 (green nodes) as two separate groups. Conservation of positions aligned with TFR2-680 is indicated by the nodes’ border colors, in which 86.7% of the sequences bear an aspartate at these positions (thin black node borders). The following most relevant identity found is for a specific group of related TFR1 sequences in which all, except for one, bear serine residues (8.8%). Other identities found in this sequence position include glycine, alanine, arginine, and proline (blue node borders); however, they are scattered around the network and are lowly represented (see the “Conservations at position 680 table” included in Figure 4b). Other sequences (2.56%) do not have an alignable residue at this position (yellow node borders) since they are truncated or have deletions at this region. This result indicates high conservation of the human TFR2-Asp680 residue among many TFR1 and TFR2 orthologous sequences. Specifically, no sequence has a tyrosine or similar residue at this position, indicating a possible evolutionary penalty for its inclusion. It seems that other strategies could have evolved in other species to bind TF by TFR1 since a large cluster of TFR1-related sequences has serine as the conserved residue at this position (Figure A2).

## 4. Discussion

In this work, we described six novel cases in five families affected by non-HFE related HH. All described patients show high levels of serum ferritin and transferrin saturation (>86%) and present the clinical phenotype of HH at an earlier age (age of diagnosis 25–37 years old) as compared with HFE-related HH. Although non-HFE related HH due to mutations on the *HFE2* gene is classically known as juvenile hemochromatosis, cases of non-HFE related HH with mutations in *TFR2* gene are also young patients (A.II.1 35 years old, A.II.2 37 years old, B.II.1 46 years old and C.II.2 25 years old). All patients reached normal iron, ferritin, and liver iron levels after intensive treatment, some of them combining the use of chelators and phlebotomies/erythroapheresis treatments.

We describe two novel mutations in the *HFE2* gene: a nonsense mutation (p.Arg63Ter) and a single-base deletion (p.Asp149ThrfsTer97) that generate a frameshift leading to a premature stop codon. Both mutations are clearly pathogenic. At the same time, we also report a new mutation in the *TFR2* gene consisting of a missense mutation (p.Asp680Tyr). Furthermore, we report a patient with a homozygous nonsense mutation (p.Gln672Ter) in the *TRF2* gene in contrast with the previously reported case where the same mutation was in heterozygous state. Finally, we describe a new case of a previously reported pathogenic mutation (p.Trp781Ter).

Due to the nature of the nonsense and frameshift mutations, their pathogenicity is obvious and supported by the clear phenotypic characteristics of the patients. All of them present the hallmarks of HH: high serum iron and ferritin levels and low levels of hepcidin, accompanied by hepatic iron overload. In the case of the missense p.Asp680Tyr mutation, the clinical features of both patients harboring this change in the TFR2 protein are the expected for the non-HFE related HH disease (high serum iron as well as the ferritin, and transferrin saturation is high). In this line, bioinformatics analysis revealed that the Asp680Tyr mutation in the TFR2 protein alters a network of interactions that preorganize the side-chain conformation of residues in TFR2 critical for its interaction with TF. This alteration is expected to destabilize the TFR2/TF binding, negatively affecting TFR2 function as a TF sensor. As a possible, although less likely, secondary effect, this mutation could also affect the ability of TFR2 to dimerize, required step for its normal activity. The combination of a reduced TF binding and a possibly reduced dimerization would result in a downregulation of the signaling cascade required for *HAMP* expression, thus resulting in reduced hepcidin production and consequently, iron accumulation. Both biochemical and clinical features are observed in both patients (A.II.1 and A.II.2), supporting the pathogenicity of the missense mutation predicted by the computational structural modelling.

Our work highlights the importance of an earlier molecular diagnosis in a specialized center to confirm the HH diagnosis, its subtype, and to prevent serious complications such as liver cirrhosis, hepatocarcinoma or cardiac dysfunction in young patients by implementing a proper and earlier iron reduction treatment.

## Figures and Tables

**Figure 1 genes-12-01980-f001:**
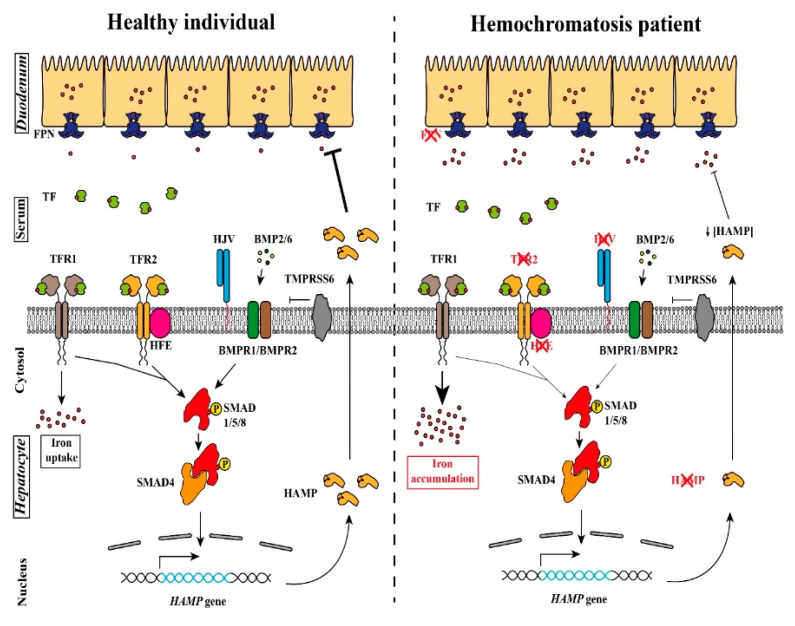
Model for the hepcidin expression and its alterations in hereditary hemochromatosis (HH). ***Left***, the coordinated signaling of the TFR1, TFR2, HFE, HJV and BMP receptors activated the SMAD complex that promotes *HAMP* expression. The hepcidin produced is then released to the blood stream and will stimulate the FPN degradation in the duodenum epithelial cells, limiting iron absorption. ***Right***, in an individual affected by HH, mutations in any of the proteins marked with a red cross result in a reduction in the serum hepcidin levels. This leads to an uncontrolled iron absorption by the duodenum that will result in an iron accumulation in the tissues.

**Figure 2 genes-12-01980-f002:**
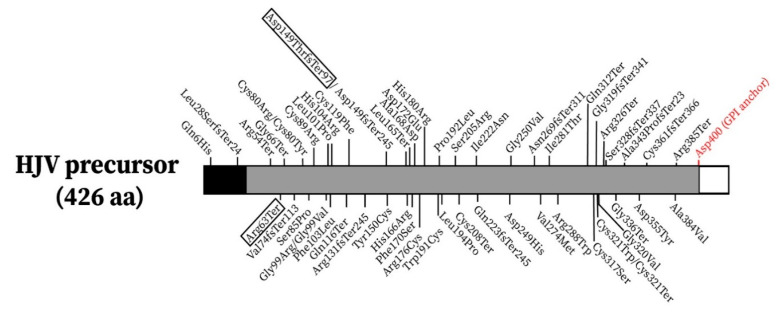
Localization of the reported pathogenic mutations in the HJV precursor. The new mutations reported in this work are boxed. The black region in the N-terminal represents the signal peptide while the white box at the C-terminal represents a portion that will be removed in the maturation process, exposing the Asp400 residue (in red) necessary for the GPI anchor of the HJV protein to the membrane.

**Figure 3 genes-12-01980-f003:**
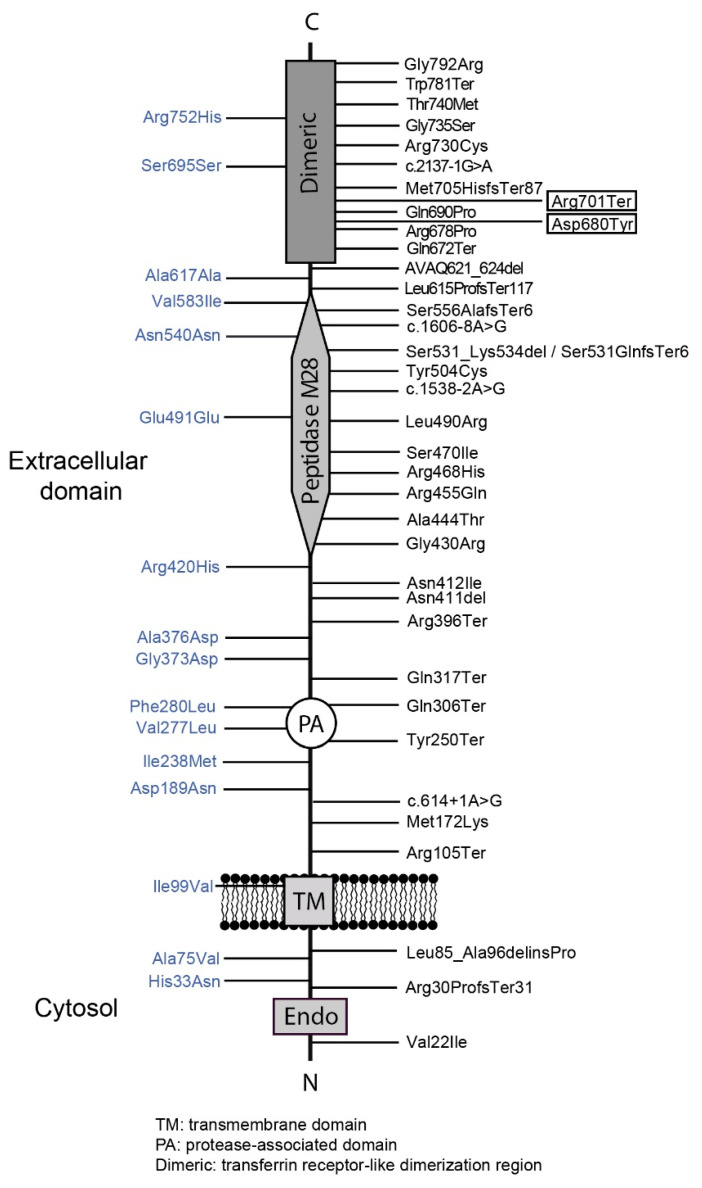
Localization of the reported mutations in the TFR2 protein. The new mutations reported are boxed. *Dimeric*, Transferrin Receptor-like dimerization region; *TM*, transmembrane domain; *PA*, Protease-Associated domain. Image adapted from Joshi et al. [23].

**Figure 4 genes-12-01980-f004:**
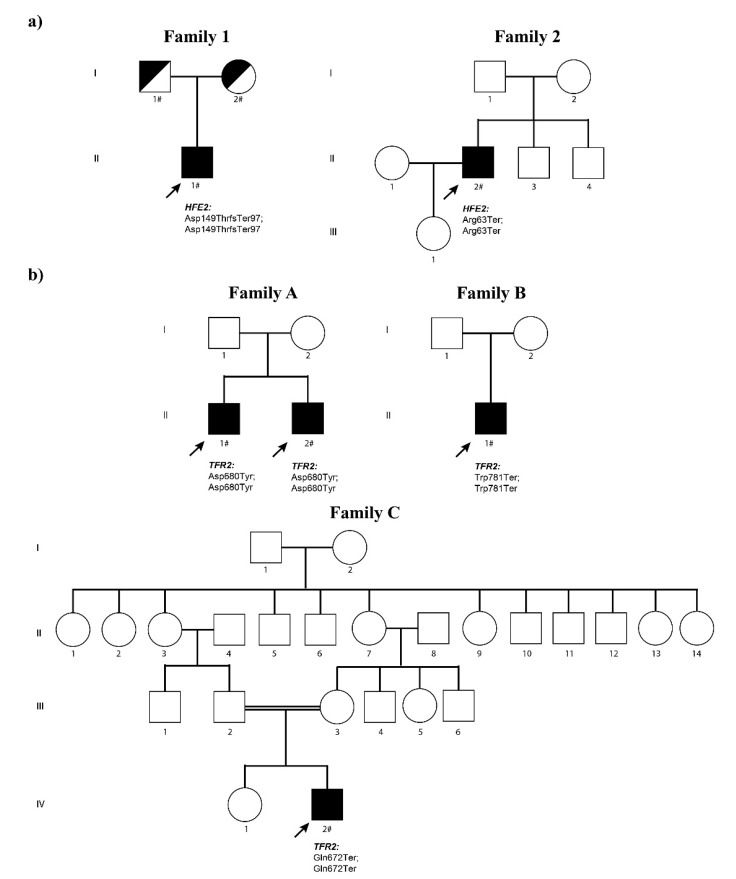
Non-HFE hereditary hemochromatosis families: families and mutations. The probands are indicated with an arrow. Black symbols denote affected individuals and half-filled black symbols unaffected carriers. Individuals studied at the molecular level are indicated with the # symbol. (**a**) Pedigrees of the two non-HFE related HH families carrying HFE2 mutations. (**b**) Pedigrees of the three non-HFE related HH families carrying TFR2 mutations.

**Figure 5 genes-12-01980-f005:**
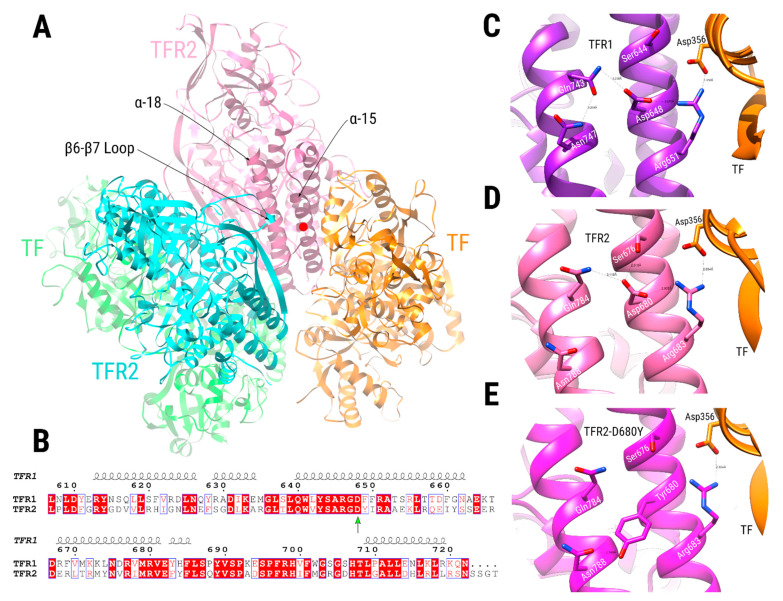
**Comparative modeling of the TFR2 dimer complexed with TF.** (**A**) TFR2 chains are shown in cyan and pink, while TF chains are shown in green and orange. Dimeric interactions between TFR2 chains, close to Asp680 position (red circle at the α-15 helix, pink chain), are made between residues in the α-18 helix of one TFR2 chain (pink chain) and the loop located between the β-6 and β-7 strands of the other (cyan chain). (**B**) Sequence alignment of the TFR1 and TFR2 proteins near the TFR2 680 position (green arrow). Numbering is according to the TFR1 sequence, and the secondary structure depiction is based on the TFR1 structure (PDB ID 1CX8). The alignment coloring was produced by the ESPript 3.0 web server [40], where position with red background means absolute sequence conservation. (**C**–**E**) Structural context of positions TFR1-Asp648 (**C**, purple), TFR2-Asp680 (**D**, pink), and TFR2-Asp680Tyr (**E**, magenta). Residue names of TFR1/2 appear in white, while residues in TF are in black. Significant interactions are shown with dashed lines.

**Table 1 genes-12-01980-t001:** Clinical, biochemical, and genetic data of patients affected of autosomal recessive *non-HFE* related HH due to *HFE2* mutations. F, female; M, male. (a) Data before iron depletion. (b) Data after iron depletion. (c) Data at first analysis. (d) Second follow-up. (e) Most recent data (May 2020). n.a., not available. * Adult serum hepcidin reference values according to (Ganz et al., 2008) are: mean of 121 ng/mL (5–95% CI: 29–254 ng/mL) for men and mean of 87 ng/mL (5–95% CI: 17–286 ng/mL) for women.

	***Family 1—Patient II.1***	***Family 2—Patient II.2***	***Normal Range***
***ex***	M	M	
***Age at clinical diagnosis***	37	34	
***Hemoglobin, Hb (g/L)***	14.6 (a)	14.9 (c)	M: 13.5–17.5
	14.7 (b)	16.7 (e)	F: 12.1–15.1
***MCV (fl)***	86.5 (a)	94.4 (c)	80–95
	84 (b)	94.4 (e)	
***Serum iron (µg/dL)***	247 (a)	267 (d)	59–158
	80 (b)	109 (e)	
***Serum ferritin (µg/L)***	4620 (a)	650 (c) 3952 (d) 27 (e)	M: 12–300
	29 (b)		F: 12–200
***Transferrin (mg/dL)***	198 (a)	218 (d)	220–400
	406 (b)	313 (e)	
***Transferrin Saturation (%)***	87 (a)	84 (d)	20–50
	14 (b)	25 (e)	
***Total Iron Binding Capacity (µg/dL)***	n.a.	301 (d)	250–400
		432 (e)	
***Hepcidin levels (ng/mL)***	n.a.	0.19	M: 29–254 *
			F: 17–286 *
***MRI liver***	457 µmol Fe/g (a)	July 2019: 47 µmol Fe/g.	<36 μmol Fe/g
	30 µmol Fe/g (b)		
	13 µmol Fe/g (2020)		
***MRI heart***	9 ms (a)	No iron overload	
	15 ms (b)		
	30 ms (2020)		
***Treatments***	Phlebotomy (2014)	Phlebotomy Desferoxamine (until March 2020)	
	Erythroapheresis		
	Desferoxamine (2014)		
	rHuEPO (2014)		
***Genetics***	HFE2: NM_213653.3:	HFE2: NM_213653.3: c.187C > T; c.187C > T p.Arg63Ter; p.Arg63Ter	
	c.445delG; c.445delG		
	p.Asp149ThrfsTer97;		
	p.Asp149ThrfsTer97		

**Table 2 genes-12-01980-t002:** Clinical, biochemical, and genetic data of patients affected of autosomal recessive *non-HFE* related HH due to *TFR2* mutations. F, Female. M, Male. (a) Data from January 2019. (b) Data from January 2018. (c) Data from November 2018. (d) Data from May 2019. n.a., not available. * Adult serum hepcidin reference values according to (Ganz et al., 2008) are: mean of 121 ng/mL (5–95% CI: 29–254 ng/mL) for men and mean of 87 ng/mL (5–95% CI: 17–286 ng/mL) for women. ^#^ Last available data from July 2021. ** Last data available from February 2021. *** Last data available from September 2019.

	*Family A—* *Patient II.1*	*Family A—* *Patient II.2*	*Family B—* *Patient II.1*	*Family C—Patient II.2*	*Normal Range*
***Sex***	M	M	M	M	
***Age at clinical*** ***diagnosis***	35	37	46	25	
***Hemoglobin, Hb (g/L)***	12.8	14.1	15 ***	13.1	M: 13.5–17.5 F: 12.1–15.1
***MCV (fl)***	n.a.	98.7	93.1 ***	n.a.	80–95
***Serum iron (µg/dL)***	244	240	282 ***	n.a.	59–158
***Serum ferritin (ng/mL)***	2000 244 **	2000 6 **	66 ***	2037 16 ^#^	M: 12–300 F: 12–200
***Transferrin (mg/dL)***	n.a.	191	241 ***	n.a.	220–400
***Transferrin Saturation (%)***	98 80 **	100 6 **	83.6 ***	94.8 4.8 ^#^	20–50
***Total Iron Binding*** ***Capacity (µg/dL)***	n.a.	242.6	n.a.	n.a.	250–400
***Hepcidin levels (ng/mL)***	0.23	0.01	n.a.	n.a.	M: 29–254 * F: 17–286 *
***MRI liver***	282.97 µmol Fe/g (a) 15.9 mg Fe/g (a)	265 µmol Fe/g (b) 14.9 mg Fe/g (b) 217.56 µmol Fe/g (c) 12.22 mg Fe/g (c) 70.33 µmol Fe/g (d) 3.95 mg Fe/g (d)	300 µmol Fe/g <36 μmol Fe/g ***	123.9 µmol Fe/g	<36 μmol Fe/g
***Treatments***	Phlebotomy Desferoxamine	Phlebotomy Desferoxamine	Erythroapheresis Phlebotomy	Phlebotomy	
***Genetics***	TFR2: NM_003227.4: c.2038G > T; c.2038G > T p.Asp680Tyr; p.Asp680Tyr	TFR2: NM_003227.4: c.2038G > T; c.2038G > T p.Asp680Tyr; p.Asp680Tyr	TFR2: NM_003227.3: c.2343G > A; c.2343G > A p.Trp781Ter; p.Trp781Ter HFE: p.Cys282Tyr	TFR2: NM_003227.4: c.2014C > T; c.2014C > T p.Gln672Ter; p.Gln672Ter	

## Data Availability

Data is available upon request.

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
