# Peer review of "New Mutations in HFE2 and TFR2 Genes Causing Non HFE-Related Hereditary Hemochromatosis"

_genes, 2021, doi:10.3390/genes12121980_

Round 1
Reviewer 1 Report
The ms reports on novel insight in non-HFE related HH, particularly regarding its presentation in Mediterranean countries. The topic of the report and the novelty of the results for this specific field are expected to be of high impact. The extensive revision of HFE2 and TFR2 reported mutation could be an important milestone for defining the state-of-the-art. I will suggest coupling molecular with clinical data of affected individuals to offer an overview of the gen-phen correlations, if available.
The clinical data of the newly reported individuals should be described in a more organized manner to make possible a comparison between individuals and families.
Please consider to revise the molecular and clinical data of the other overlapping non-HFE related HH.
Author Response
We thank reviewers’ comments to improve our manuscript. We have addressed in a point-by point revision all comments suggested by the reviewers. See below
Reviewer #1
The ms reports on novel insight in non-HFE related HH, particularly regarding its presentation in Mediterranean countries. The topic of the report and the novelty of the results for this specific field are expected to be of high impact. The extensive revision of HFE2 and TFR2 reported mutation could be an important milestone for defining the state-of-the-art. I will suggest coupling molecular with clinical data of affected individuals to offer an overview of the gen-phen correlations, if available.
Clinical and genotype data are now? available together in tables 1 and 2.
The clinical data of the newly reported individuals should be described in a more organized manner to make possible a comparison between individuals and families.
We agree with the suggestion of the reviewer, and we have now reorganized our data. The clinical data of the patients is available side-by-side in both tables 1 and 2.
Please consider to revise the molecular and clinical data of the other overlapping non-HFE related HH.
Thanks for this suggestions. In the introduction section where we define at the molecular level all types of HH, we have now included the below sentence where we compare at the clinical level the different types of HH (lanes 98-99).
“Clinically, HH types 2a, 2b and 3 present with a more severe and earlier iron overload complications compared with type 1 and 4b.”

Reviewer 2 Report
The study brings new data on molecular background of genetic iron overload.
Critical remarks:
Section references must be corrected as there are no full names of the citated papers' authors.
In many places of the text number of a figure 3 (description of probands) should be checked to 4.
Please confirm that only one of the patients was a carrier of HFE gene mutation (B.II.1), in other cases HFE gene mutations were excluded
268-269 line – please confirm that also B.II.1 presented iron overload in MR imaging.
The patient 1.II.1 is not described satisfactorily: "magnetic resonance shows severe liver and cardiac iron overload" - what were the values of iron assessment? And how to explain a lack of cardiac dysfunction if cardiac iron overload was severe? Usually patients with HH diagnosis (I mean HFE-HH which is the most frequent one) have a good tolerance of treatment with phlebotomies - this patient needed recombinant human EPO, he was treated also with iron chelator. Please explain if any hematological investigation has been performed to exclude hematopoietic system disease as this pathology could be responsible for severe iron loading not the genetic dysfunction of the HFE2 gene. If it was not done - such explanations and doubts shoul be added to section discussion.
From the clinical point of view, the information about combination treatment with phlebotomies and iron chelation is valuable but for clinicians it is important how lack of efficacy of phlebotomies was defined. Do authors want to underline the severity of iron overload or resistance to phlebotomies in carriers of non-HFE HH. Is there any reason for disturbed erythropoiesis in this genetic dysfunction?
Please put proper data on type of iron chelator which was used in the teratment as in the text it is written that patients were treated with desferoxamine and in a table 2 defiraserox ??(deferasirox is an oral drug) is mentioned.
Finally a short comment to the conclusion
“Our work highlights the importance of an earlier molecular diagnosis in a specialized center to confirm the HH diagnosis, its subtype and prevent serious complications such as liver cirrhosis, hepatocarcinoma or cardiac dysfunction in young patients by implementing a proper and earlier iron reduction treatment” – I partially agree with this statement because from the clinical point of view the most important is to decide if the patient needs treatment or not or how deep the abnormalities of iron status are in a single clinical case. The genetic background makes it better to understand the pathophysiology of iron overload but in fact, in the majority of clinical situations there is no access to advanced molecular techniques (usually due to high cost) which enable complete molecular diagnosis.
Genetic testing is not offered to persons who do not present clinical symptoms of iron overload. Unless the authors have a proposed algorithm
Author Response
We thank reviewers’ comments to improve our manuscript. We have addressed in a point-by point revision all comments suggested by the reviewers. See below
Reviewer #2
The study brings new data on molecular background of genetic iron overload.
Critical remarks:
Section references must be corrected as there are no full names of the citated papers' authors.
References have been corrected to be shown with the proper format.
In many places of the text number of a figure 3 (description of probands) should be checked to 4.
Thanks for detecting this mistake. The correct figure number has been added to the text.
Please confirm that only one of the patients was a carrier of HFE gene mutation (B.II.1), in other cases HFE gene mutations were excluded
Yes, these cases are non-HFE hemochromatosis patients and none of them is a HFE C282Y homozygous patient, what will classify it as a HFE Hemochromatosis classical patient. Patient B.II.1 is heterozygous for the C282Y mutation in the HFE gene. That genotype by itself does not classify the patient as a hemochromatosis patient. We have added a statement of the HH-related gene mutation status for all the patients when available:
“HFE testing for Cys282Tyr and His63Asp variants was negative.” Lanes 270-271
“No other variants of interest were found in any of the HH-related genes.” Lane 281
“Both probands are homozygous WT for the His63Asp, Ser65Cys and Cys282Tyr HFE mutations.” Lane 292
“No other mutations in HH-related genes were found.” Lane 307
All patients have been studied at the level of genetic sequencing for all the Hemochromatosis genes by NGS.
268-269 line – please confirm that also B.II.1 presented iron overload in MR imaging.
The MRI detection of the iron overload in patient B.II.1 was stated in lines 196-197: “severe hepatic iron accumulation (300 µmol Fe/g) detected by hepatic magnetic resonance.” And it is also reported in Table 2.
The patient 1.II.1 is not described satisfactorily: "magnetic resonance shows severe liver and cardiac iron overload" - what were the values of iron assessment?
We now better detailed patient’s description on page 4, lines 141-169:
“Patient 1.II.1 (Figure 4a left) is a male diagnosed at the age of 37. Biochemical data revealed mild basal fasting hyperglycemia with mild insulin resistance (HOMA index of 3.7), vitamin D and folic acid deficit and hypogonadotropic hypogonadism. Hepatic ultrasound showed a slightly increased liver with irregular structure, elastography revealed liver fibrosis and magnetic resonance shows severe liver [T2* 1 msec, corresponding to a liver iron concentration (LIC) of 457 µmol/g dry weight[27] and cardiac (T2* 9 msec) iron overload. Liver biopsy showed cirrhosis and marked iron overload (Figure A1). Despite cardiac overload, echocardiography showed normal morphology and ventricular function with ground glass aspect of the medio-basal ventricular septum. This indicates that T2* was identifying preclinical iron deposition as previously observed in patients with transfusion-dependent thalassemia with cardiac iron overload[28, 29]. Because of the severe iron overload, the patient underwent to combined therapy to remove as soon as possible the systemic iron burden. Based on previous experience we opted for erythroapheresis to maintain isovolemia (considering the cardiac iron overload) plus recombinant erythropoietin administration (8000 UI/week for three weeks/month) to maximize the efficacy of erythroapheresis[30]. In addition, subcutaneous deferoxamine infusion (1 g twice/day for 5 days/week) was used. Iron depletion was achieved after the removal of 15 g of iron in one year. Afterwards, phlebotomy therapy every 3-4 months was done maintaining ferritin levels around 50 ng/mL.”
And how to explain a lack of cardiac dysfunction if cardiac iron overload was severe?
Despite the presence of cardiac iron overload in juvenile hemochromatosis patients, previous studies showed that only 35-40% of these patients develop cardiac dysfunction (Br J Haematol 2002;117:973; Orphanet J Rare Dis 2019;14:171). As well documented in thalassemia major patients with transfusional iron overload, iron accumulation precedes the development of organ dysfunction in the heart (Wood et al Ann N Y Acad Sci. 2005;1054:386; Eur Heart J 2001;22:2171-2179). This observation can be also applied to other organs (liver, pancreas) and to other causes of iron overload including hemochromatosis (Am J Gastroenterol 2005;100:837; Am J Gastroenterol 2001;96:567; Blood 2000;343:327). We now added a sentence in the text (page 4, lines 141-169).
Usually patients with HH diagnosis (I mean HFE-HH which is the most frequent one) have a good tolerance of treatment with phlebotomies - this patient needed recombinant human EPO; he was treated also with iron chelator. Please explain if any hematological investigation has been performed to exclude hematopoietic system disease as this pathology could be responsible for severe iron loading not the genetic dysfunction of the HFE2 gene. If it was not done - such explanations and doubts should be added to section discussion.
In concrete in this patient due to the presence of severe iron overload, he underwent a combined therapy to remove as soon as possible the systemic iron burden. Based on previous experience we opted for erythroapheresis to maintain isovolemia (considering the cardiac iron overload) plus recombinant erythropoietin (rHuEPO) administration 8000 UI/week for three weeks/month to maximize the efficiency of erythroapheresis (Haematologica 2005; 90:717-718). In addition, subcutaneous deferoxamine infusion (1 g/day for 5 days/week) was used. No haematological disorders were present (Baseline Hb level 14.6 g/dL, MCV 86.5 fl, normal leucocytes 6.65x103/ µL and platelets number 168x103/µL) in this patient. We now added these details in the text (page 4, lines 141-169).
From the clinical point of view, the information about combination treatment with phlebotomies and iron chelation is valuable but for clinicians it is important how lack of efficacy of phlebotomies was defined. Do authors want to underline the severity of iron overload or resistance to phlebotomies in carriers of non-HFE HH.
Lack of efficiency of phlebotomies is defined as fast anemization by a normal phlebotomy of 500 ml, what is normally not seen in HFE and non-HFE patients. While patients with ferroportin disease, previously known as HH type 4a and nowadays considered a different disease than Hereditary Hemochromatosis Type 4b, have been reported to be intolerant to strong phlebotomies treatment this is less frequent in HFE and non-HFE HH patients. This less frequency of intolerance to phlebotomies is confirmed by the fact that all but one (2.II.2) of our six non-HH patients are tolerant to a normal schedule of phlebotomies (i.e. normally 500 ml are removed weekly).
In severe iron overloaded hereditary hemochromatosis patients, it is usually done a combined dual treatment by phlebotomies together with iron chelation, for a more effective treatment and a faster removal of liver iron overload.
Is there any reason for disturbed erythropoiesis in this genetic dysfunction?
In HH there is not a direct erythropoiesis dysfunction since patients are not anaemic. If we consider non-HFE HH as the newly classification reported in (Girelli et al. Blood 2021, PMID: 34601591) experts conclude to define HH as a non-hematological disease without erythropoiesis implications.
Please put proper data on type of iron chelator which was used in the treatment as in the text it is written that patients were treated with desferoxamine and in a table 2 defiraserox ??(deferasirox is an oral drug) is mentioned.
Thanks for detecting this mistake, we realise that the Table 2 reference to treatment with Deferasirox (was wrong misspelled as defiraserox) and indeed it was wrong reported. We added the correct treatment that is treatment with desferoxamine.
Finally a short comment to the conclusion
“Our work highlights the importance of an earlier molecular diagnosis in a specialized centre to confirm the HH diagnosis, its subtype and prevent serious complications such as liver cirrhosis, hepatocarcinoma or cardiac dysfunction in young patients by implementing a proper and earlier iron reduction treatment” – I partially agree with this statement because from the clinical point of view the most important is to decide if the patient needs treatment or not or how deep the abnormalities of iron status are in a single clinical case. The genetic background makes it better to understand the pathophysiology of iron overload but in fact, in the majority of clinical situations there is no access to advanced molecular techniques (usually due to high cost) which enable complete molecular diagnosis.
Genetic testing is not offered to persons who do not present clinical symptoms of iron overload. Unless the authors have a proposed algorithm
We totally agree with the reviewer comment. For an update on HH algorithm it is very helpful to check the one recently proposed by Girelli et al. Blood 2021, PMID: 34601591. We did some contribution to that excellent work.
